# Diagnostic Accuracy and economic value of a Tiered Assessment for Fetal Alcohol Spectrum Disorder (DATAforFASD): Protocol

Dianne C Shanley [iD],[1,2] Melanie Zimmer-Gembeck,[1,2] Amanda J Wheeler [iD],[1,3] Joshua Byrnes,[1,4] Robert S Ware [iD],[1] Wei Liu,[1] Gabrielle Simcock,[1] Codi White,[1] Sarah Horton,[5] Marjad Page,[1,6] Doug Shelton,[1,4,7] Haydn Till,[1,4,7] Ianthe Mills,[8] Carly Hislop,[9] Katrina Harris,[10,11] Alison Crichton,[10,11] Natasha Reid,[12] Sheena Reilly,[1] Karen Moritz,[13] Kerryann Walsh,[14] Sharyn Rundle-Thiele,[15] Erinn Hawkins[1,2]

**Correspondence to**
Professor Dianne C Shanley;
d.shanley@griffith.edu.au

## ABSTRACT

**Introduction** Australian practices for diagnosing fetal alcohol spectrum disorder (FASD) are lengthy and require specialist expertise. Specialist teams are based in urban locations; they are expensive and have prolonged waitlists. Innovative, flexible solutions are needed to ensure First Nations children living in rural/remote communities have culturally appropriate and equitable access to timely diagnosis and support. This study compares the accuracy of rapid assessments (index tests) that can be administered by a range of primary healthcare practitioners to specialist standardised FASD assessments (reference tests). The cost-efficiency of index tests will be compared with reference tests.

**Methods and analysis** At least 200 children aged 6–16 years at-risk of FASD will be recruited across at least seven study sites. Following standards for reporting diagnostic accuracy study (STARD) guidelines, all children will complete index and reference tests. Diagnostic accuracy statistics (including receiver operating curves, sensitivity, specificity, positive and negative predictive values and likelihood ratios) will identify whether rapid assessments can accurately identify: (1) the presence of an FASD diagnosis and (2) impairment in each neurodevelopmental domain, compared to comprehensive assessments. Direct and indirect healthcare costs for index tests compared to reference tests will be collected in primary healthcare and specialist settings.

**Ethics and dissemination of results** Children's Health Queensland Hospital and Health Service Human Research Ethics Committee (HREC/20/QCHQ/63173); Griffith University Human Research Ethics Committee (2020/743). Results will assist in validating the use of index tests as part of a tiered neurodevelopmental assessment process that was co-designed with First Nations community and primary healthcare practitioners. Outcomes will be summarised and provided to participating practitioners and sites, and disseminated to community health services and consumers. Findings will be presented at national and international conferences and published in peer-reviewed journals.

**Trial registration number** ACTRN12622000498796.

## INTRODUCTION

Fetal alcohol spectrum disorder (FASD) affects between 1 - 5% of Australian children, and up to 19% in some remote Australian communities, which is higher than any other neurodevelopmental disorder.[1–4] To identify FASD, current Australian guidelines recommend comprehensive diagnostic assessments (reference tests) across 10 neurodevelopmental domains, delivered by multidisciplinary teams. These assessments are primarily available in Australia at specialist services, making this a lengthier and more costly diagnostic process than any other neurodevelopmental disorder.[5] Current waitlists for Australian children suspected of FASD are up to 3 years at some health services. Overall, specialist services are only seeing a small proportion of suspected cases and children from rural and remote regions are left behind.[6 7]

To meet the demand for diagnosis, our Australian health system either needs an inordinate number of specialists and multidisciplinary clinics, or we need to think differently about how to diagnose FASD. Our health system is unlikely to be able to resource the longest, most costly assessment process for one of the most prevalent neurodevelopmental disorders in children, but the use of valid and reliable rapid assessment tools could leverage capacity within primary care, increasing vulnerable children's access to diagnosis.

In this project we will conduct a two-phase study to: (1) determine the diagnostic accuracy of rapid neurodevelopment assessments (index tests) that can be used to detect FASD

BMJ

## STRENGTHS AND LIMITATIONS OF THIS STUDY

⇒ DATAforFASD is the first large, prospective, multi-site trial to compare comprehensive assessments to rapid assessments across nine domains of neurodevelopment so that we can better understand the accuracy of using rapid assessments to screen and diagnose neurodevelopmental disorders like fetal alcohol spectrum disorder (FASD).

⇒ A sample size of at least 200 children is uncommon in FASD research and may prove to be challenging, because completing gold-standard comprehensive assessments on nine neurodevelopmental domains requires around 2 days of multidisciplinary specialist (paediatrician/allied health) time per child.

⇒ The rapid assessment fits within a co-designed tiered assessment process that is responsive to cultural protocols and limited healthcare resources in remote regions. It includes the Rapid Neurodevelopmental Assessment, an observational functional neurodevelopmental assessment and the Behavioural Assessment Scale for Children, Third Edition.

⇒ This project follows standards for reporting diagnostic accuracy study (STARD) guidelines, providing the methodological rigour necessary to assess the diagnostic accuracy of rapid assessments in identifying FASD, as well as their accuracy in identifying neurodevelopmental impairment across nine domains; the cost benefit of using rapid assessment will be explored for families and health services.

⇒ Establishing valid, accurate rapid assessments will enable: (a) early detection of at-risk children, particularly in rural and remote communities where access to care is inequitable; (b) non-specialist care providers (health, education, youth justice, child protection) to confidently contribute to information needed for diagnostic purposes; (c) enhanced triage, resulting in better quality referrals to specialists; (d) the use of rapid assessments for diagnostic purposes, potentially addressing a systemic healthcare challenge by offsetting the longest most costly assessment process for the most prevalent preventable neurodevelopmental disorder.

in primary healthcare (Phase I)[8]; and (2) examine the economic benefits of leveraging primary healthcare to support specialist practitioners in the diagnosis of FASD, in terms of cost for families and healthcare systems (Phase II).

## METHODS AND ANALYSIS
### Phase I: diagnostic accuracy
### Study design

A prospective diagnostic accuracy study [following standards for reporting diagnostic accuracy study (STARD) guidelines] will compare index tests to reference tests for 9 of 10 neurodevelopmental domains commonly used by specialist, multidisciplinary teams when diagnosing FASD.[9] The brain structure domain was not included because its' reference test was already rapid (ie, history of seizures, head circumference or brain imaging if available). Therefore, no index test for this domain was needed. Information for this domain will still be collected for diagnostic purposes.

### Primary and secondary outcomes

The first primary outcome will determine the diagnostic accuracy of identifying FASD with index tests compared with reference tests. The second primary outcome will determine the accuracy of identifying impairment within each domain assessed. It is hypothesised that the rapid assessments (index tests) will have good sensitivity and specificity when identifying: (a) the presence of an FASD diagnosis (i.e., presence of three severe neurodevelopmental concerns plus prenatal alcohol exposure), and (b) impairment in each neurodevelopmental domain.

Comprehensive neurodevelopmental assessments can result in the accurate diagnosis of multiple neurodevelopmental disorders. Therefore, a secondary outcome will determine the accuracy of index tests in diagnosing other childhood neurodevelopmental disorders that co-occur, or occur independently of FASD, such as attention-deficit hyperactivity disorder (ADHD) and autism spectrum disorders (ASD).

### Study sites and participants

At least 200 children referred to one of seven partner organisations conducting neurodevelopmental assessments will complete index tests and reference tests between April 2022 and April 2025. Sites include the Gold Coast Hospital and Health Service, Sunshine Coast Hospital and Health Service, North West Hospital and Health Service, Townsville Hospital and Health Service, Monash Children's Hospital, Gidgee Healing and Kambu Aboriginal and Torres Strait Islander Corporation for Health. Additional sites may commence to ensure the projected sample size is reached. Inclusion criteria are:

▶ 6–16 years-old;
▶ suspected or confirmed prenatal alcohol exposure upon referral;
▶ able to speak English.

There are no exclusion criteria. Sites include publicly and privately funded clinics (e.g., Aboriginal Community Controlled Health Services, state-funded hospital outpatient clinics and university-based outpatient clinics). Site catchment areas cover remote, rural and metropolitan children. Annually, the proportion of First Nations children who attend sites for a neurodevelopmental assessment range from 20% to 90%.

### Sample size

Sites conduct between 1-4 neurodevelopmental assessments per month as part of usual care. The number of FASD diagnoses completed at sites in the previous year was used to project a total recruitment sample of 288 children over 3 years of data collection. Allowing for 30% attrition, a final sample of 200 participants with 50% representation from rural and remote sites and 50% representation of First Nations children will be possible.

The probability of FASD diagnosis in this sample will be moderate (estimated at 50–60% across sites) because children are referred to sites due to significant developmental delays and suspected prenatal alcohol exposure. Two hundred participants provide a large enough sample for precise estimates of sensitivity and specificity. For example, assuming 55% of the sample have FASD, and the true sensitivity and specificity is 80% and 90%,

respectively, we will be able to estimate sensitivity to within ±7.5% and specificity to within ±6.2% (alpha=0.05).

## Measures

### Participant information

*Demographic information* about the family (family composition, parents' education level and employment status) and about the child (sex, age, date of birth and cultural background) will be collected.

### Clinical assessment information

*Child developmental history* will be obtained from caregiver interviews and will include: gestational age at birth, prenatal exposures (e.g., prescription medication, illicit drug use), birth complications, family and child medical and mental health history, children's adverse life events, and parent reports of children's learning, speech, behavioural and emotional difficulties. Where possible, *maternal alcohol consumption in pregnancy* will be assessed using the three-item Alcohol Use Disorders Identification Test-Concise (AUDIT-C).[10] Each item is scored from 0 to 4, with higher scores indicating increased alcohol usage. Total scores range from 0 (no alcohol consumption) to 12; a score of 3 or above for women indicates hazardous drinking. The AUDIT-C has convergent validity with the 10-item version of the scale.[11] *Sentinel features,* including the standard frontal, three-quarter and lateral facial photographs will be collected and analysed using the Fetal Alcohol Spectrum Facial Photographic Analysis Software V.2.0.[12]

*Index tests* were selected by clinical specialists with expertise in FASD, subsequently approved by site practitioners and the project steering committee. Table 1 outlines the rationale for index test selection.

*The Rapid Neurodevelopmental Assessment (RNDA)* is a 30-minute observational and functional assessment identifying neurodevelopmental impairment in children from birth to 16 years. As well as screening for vision and hearing problems (that can affect task performance), it assesses motor (gross and fine), language, cognition, memory, attention, behaviour, and self-care. A series of age-appropriate structured tasks delivered by trained administrators follow specified decision rules detailed in the administration and scoring manual.[13] As a functional assessment (rather than a normed assessment), children are given prompts and modelling to complete tasks if needed. Degree of impairment is classified as: typical, mild, moderate or severe impairment. Research demonstrates that the RNDA has sound to excellent inter-rater reliability across the neurodevelopmental domains with infant, child and adolescent populations.[14–16] The RNDA has been reviewed by First Nations health practitioners and administered with First Nations children.[17] *The Behavioural Assessment Scale for Children, Third Edition* is a 20-minute parent- and teacher-report of 2- to 21-year-old behavioural, emotional and social skills, as well as adaptive, attention and executive functioning in home, school and community settings.[18] Both parent-report and teacher-report forms have excellent internal validity, very good test–retest reliability and inter-rater reliability.[18]

*Reference tests* were determined over a 6-month cross-site consultation process. Agreement on reliable and valid measures for nine domains was achieved in six steps (see table 2).

Table 3 outlines how the index tests correspond to reference tests selected for each neurodevelopmental domain used to diagnose FASD (excluding brain structure). All

| Table 1 | Rationale for selection of index tests |
| --- | --- |
| **Rapid Neurodevelopmental Assessment[13]** | **Behaviour Assessment System for Children Third Edition[18]** |
| Delivered by non-specialists trained to certification in administration, important for rural and remote sites lacking specialists. | Established reliability and validity within the Australian context[25] |
| Developed for use in low social-economic status populations,[26][27] important for rural and remote Australians where 13.3% of the population live below the poverty line.[28] | Covers a broad array of neurodevelopmental domains, including executive and adaptive function, key fetal alcohol spectrum disorder diagnostic domains not commonly assessed in other parent-report measures of child psychosocial functioning[29] |
| Appropriate for children 0–17 years old, not limited to young children | |
| Utilises independent, direct, observational assessment of neurodevelopment, not solely reliant on parent-report | |
| Categorises impairment according to severity (typical, mild, moderate, severe), not dichotomous (e.g., the presence or absence of impairment) | |
| Instructions can be non-verbal (shown not spoken), important for culturally diverse populations and children with English as a second language | |

**Table 2** Steps to establish cross-site agreement for reference tests

| Step | Activity |
|---|---|
| 1 | An international survey of clinical practice broadly ascertained the standard assessment protocol for each domain at neurodevelopmental assessment clinics globally. |
| 2 | A systematic review outlined the standard assessment protocol for each domain when detecting impairment for a fetal alcohol spectrum disorder diagnosis in internationally published research.[30] |
| 3 | Identification of tools from Step 1 and 2 with established Australian norms. |
| 4 | A presentation to lead practitioners at each site introducing outcomes of Steps 1, 2 and 3. Lead practitioners took information to their clinical team to compare with site specific assessment protocols for each domain. |
| 5 | Monthly group meetings with lead practitioners across sites discussed pros and cons of preferred reference tests, until consensus was achieved. |
| 6 | When consensus could not be achieved on domain tests, the underlying construct of the domain was defined and consensus was achieved on subtests measuring similar constructs. For example, for academic achievement, a valid and reliable reading subtest exists in both the Wide Range Achievement Test[31] and the Wechsler Individual Achievement Test.[32] |

reference tests align with recommendations from the Australian guide to the diagnosis of FASD.[5]

### Procedure

Practitioners at each site will ask families with children who meet inclusion criteria if they consent to participate. Each site will administer index and reference tests as clinically appropriate. Index tests will be administered by trained practitioners including Aboriginal Health Workers, nurses, midwives, allied health, General Practitioners (GPs) and/or paediatricians (when available). Reference tests will be administered by allied health clinicians with relevant qualifications and credentialing for the tool (e.g., psychologists, occupational therapists, speech therapists), who have training and experience diagnosing FASD, consistent with the Australian guide to the diagnosis of FASD.[5] Where possible, index tests

**Table 3** Index and reference tests used to assess neurodevelopmental domains

| Domain | Index test | Reference test |
|---|---|---|
| Motor | RNDA: Gross Motor and Fine Motor subscale | M-ABC 2 or BOT-2 |
| Cognition | RNDA: Cognition subscale | WISC-V |
| Language | RNDA: Language subscale; BASC-3: Functional Communication scale | CELF-5* |
| Academic achievement | BASC-3: Learning Problems scale | WIAT-III or WRAT-5 or WJ-IV |
| Memory | RNDA: Memory subscale | CMS: Stories and Dot Location subscales |
| Attention | RNDA: Behaviour subscale (attention items); BASC-3: Attention Problems clinical scale | Connors-3 or Conners-4 |
| Executive function | BASC-3: Executive Functioning index score | BRIEF-2 |
| Affect regulation | RNDA: Behaviour subscale (affective items); BASC-3: Anxiety and Depression clinical Scale | Previous diagnostic status (depression/anxiety) |
| Adaptive behaviour, social skills Social communication | RNDA: Self Care subscale; RNDA: Behaviour subscale (peer play item); BASC-3: Adaptive Skills composite | ABAS-3 or Vineland-3 |

Note: Rapid Neurodevelopmental Assessment: Australian Edition (RNDA)[13]; Behaviour Assessment System for Children, Third Edition (BASC-3)[18] ; Movement Assessment Battery for Children: Second Edition (M-ABC 2)[33]; Bruininks-Oseretsky Test of Motor Proficiency, Second Edition (BOT-2)[34]; Wechsler Intelligence Scale for Children, Fifth Edition (WISC-V),[35] Australian and New Zealand Standardised Edition; Clinical Evaluation of Language Fundamentals, Fifth Edition (CELF-5)[36]; Wechsler Individual Achievement Test, Third Edition (WIAT-III)[32]; Wide Range Achievement Test, Fifth Edition (WRAT-5)[31]; Woodcock-Johnson Achievement Test, Fourth Edition, Australasian Adaptation (WJ-IV)[37]; Children's Memory Scale (CMS)[38]; Connors, Third Edition (Connors-3)[39] or Connors, Fourth Edition (Connors-4)[40]; Behaviour Rating Inventory of Executive Function, Second Edition (BRIEF-2)[41]; Adaptive Behaviour Assessment System, Third Edition (ABAS-3)[42]; Vineland Adaptive Behaviour Scales, Third Edition (Vineland-3)[43].
*Equivalent language assessment validated for a First Nations population.

and reference tests will be administered by different practitioners. Practitioners administering index tests will not have access to reference test results, and vice versa, until all tests are scored. When index or reference tests have been completed by third parties (e.g., the school), families will be asked for their consent to share data with participating study sites. A member of the research team will audit health records to extract test scores from completed assessments.

All practitioners and the research team will receive training in study protocols to ensure fidelity prior to administering tests and extracting data. Training will include: purpose of the project, rationale for test selection and use of the RNDA. RNDA training includes studying standardised training videos developed by certified RNDA trainers, scoring training tapes, video recording assessments with a sample child and a review of this video by a certified scorer (to ensure fidelity). If a site does not have practitioners available to complete a test, clinicians with relevant qualifications from the research team will administer the missing test. Check-in meetings with practitioners at each site will be held quarterly to ensure continued fidelity with study protocols. All index and reference tests aim to be completed within 9 months of the administration of the first test.

## Phase II: economic analysis
### Study design
A cost-efficiency analysis will determine the cost and potential economic net benefits of: (a) implementing the rapid index tests compared with comprehensive reference tests, and (b) conducting assessments in primary healthcare compared with specialist services.

### Primary and secondary outcomes
Economic analyses will be secondary outcomes, which include: the (1) direct healthcare costs, and (2) indirect costs of diagnosing FASD using index tests compared with reference tests, (3) the direct healthcare costs and (4) indirect cost savings of delivering tests in primary healthcare compared with specialist clinics.

### Participants
Two hundred families enrolled in Phase I will be asked about caregiver costs associated with index tests and reference tests. All sites will record health service costs associated with index tests and reference tests. Sites conducting comprehensive reference FASD assessments prior to the study will record health service costs of usual care for FASD assessments retrospectively.

### Measures
#### Caregiver costs
Direct and indirect assessment costs for each family including clinical fees, childcare, accommodation and meals for assessment days will be surveyed from caregivers.

#### Health service costs
Prospective cost of delivering *index and reference tests* to participating children will be surveyed from practitioners and include: (1) discipline of the practitioner administering the test; (2) mode of delivery (e.g., clinic, home, school or telehealth); (3) travel time and costs; (4) test administration time; and (5) indirect administration time for the assessment (e.g., scoring and interpretation).

Usual cost of *reference tests* will be collected from clinical managers in a structured interview, including time taken to administer and score each measure, travel time and other costs.

### Procedure
Prior to participant recruitment, the cost of index and reference tests (usual care) for health services will be retrospectively collected in a structured phone/video interview from site managers. Site manager responses will be recorded, transcribed and data will be extracted. Expected completion time for index and reference tests will also be extracted from test manuals. Families will be recruited and enrolled in the study as outlined in Phase I. Enrolled families and healthcare practitioners will complete surveys with one member from the research team that capture prospective cost information at the completion of testing. Therefore, the cost of index and reference tests will be collected retrospectively for tests administered as part of prior usual care from health service interviews, and costs will be collected prospectively for patients and health services recruited during the study period.

### Patient and public involvement
The rapid assessment used in the current project forms part of a tiered approach to screening, diagnosing and supporting FASD in remote Australian communities that was co-designed with First Nations Elders. Co-design involved First Nations Elders, a community advisory group, First Nations remote health practitioners, clinical experts and researchers. The research questions in the current project originated from a community advisory group because they asked, 'What is necessary and sufficient to screen for and diagnose FASD?' (ensuring judicious investment of resources), and 'Who in the health care system is best placed to diagnose FASD in a remote setting?' (reducing bottlenecks in specialist settings). Ongoing collaboration (focus groups, informal yarning, 3-monthly 1-week visits to community) between this group and the research team supported the development of the current protocol and this group has agreed to continue support for the duration of the project. Practitioners at new partner sites will be asked to identify a community representative interested in joining our advisory group. The advisory group will be informed of project progress and at the end of data collection, they will be involved in interpreting study results and will support dissemination of findings to local communities and the general public.

## Data analysis
### Phase I: diagnostic accuracy

All data will be entered into an Research Electronic Data Capture (REDCap) database hosted by Griffith University.[19] Data will be identifiable until all test results and survey data are matched to participants. Data will be de-identified for analysis. Prespecified test positivity cut-offs reflecting clinical norms (e.g., T score>70) will be used, receiver operating curves will also be used to explore best possible cut-offs. Standard diagnostic statistics will be calculated, including: receiver operating curves, sensitivity, specificity, positive and negative predictive values and likelihood ratios (using 95% confidence intervals).[20] Kappa statistics will be used to compare the reliability of the classification of children as normal/moderately impaired versus severe on each measure of the neurodevelopmental domains.[21] Continuous scores are available for many of the assessments, which will be used to calculate correlations between matched scores from the index tests and reference tests. Bland-Altman plots will examine the agreement between matched index test and reference test scores, overall and by domain.[21] These plots will also help to determine if any difference between the index and the reference is influenced by the actual scores on the tests (e.g., if the index-standard difference is larger when children have low or high scores). Diagnostic outcomes from the index tests will be compared with diagnostic outcomes from the reference tests to explore whether the index tests function as an effective screening tool for FASD.

### Phase II: economic analysis

A multivariable generalised linear model of the cost of diagnosis, with the predictive variable as the index test or reference test, will estimate the incremental cost between the two alternative diagnostic processes adjusting for differences in patient geographical location and other potential confounders. Subgroup cost analysis will explore heterogeneity of results between patients' subgroups by including an interaction term (e.g., location, diagnostic procedure) within the multivariable generalised linear model of cost. The expected value of implementation will be calculated as the net monetary benefit of the index tests (i.e., monetary benefits — costs) multiplied by the population of patients expected to benefit from the reference tests. Probabilistic and one-way sensitivity analysis will be used to characterise the uncertainty in the economic results, highlighting the most important drivers for the value of implementing the index tests.

### Summary

This research aims to establish the accuracy and cost-efficiency of rapid, culturally-responsive screening and assessment tools, which are part of a co-designed, tiered neurodevelopmental assessment process for diagnosing FASD in primary healthcare.[17 22–24] Diagnosing children with FASD in primary healthcare will lessen the burden on specialist systems and help to place developmentally and socially/emotionally vulnerable children on support trajectories earlier. Our findings will help to demonstrate how primary healthcare, which is readily available to families regardless of their location, can support the assessment of FASD. Even partially adding information available to specialists can streamline subsequent interventions. Providing primary healthcare practitioners with the knowledge and tools to quickly identify children who are developmentally at-risk, to understand when to diagnose children and when to seek specialist involvement, will reduce the demand on overburdened specialist clinics. This will assist in providing timely access for assessment and treatment of children suspected of having FASD in Australia.

## ETHICS AND DISSEMINATION
### Ethics

The study has received ethical approval from the Griffith University Human Research Ethics Committee (2020/743) and Children's Health Queensland Hospital and Health Service Human Research Ethics Committee (HREC/20/QCHQ/63173). Consent to conduct research was obtained from the Kalkadoon Native Title Aboriginal Corporation Registered Native Title Bodies Corporate, representing the traditional owners of the lands from the remote data collection site.

### Dissemination of findings

All study data will be archived on a secure server at Griffith University for at least 5 years after the last publication. Re-identifiable participant data will be collected on a secure web-based platform (REDCap) and exported to Griffith University servers for archiving. A summary report will be disseminated to participating site practitioners and partner organisations. The community advisory group will support dissemination of outcomes within local communities. Outcomes of the project will be disseminated widely at national and international conferences, published in peer-reviewed journals and disseminated on social media.

**Author affiliations**
[1]Menzies Health Institute of Queensland, Griffith University, Gold Coast, Queensland, Australia
[2]School of Applied Psychology, Griffith University, Gold Coast, Queensland, Australia
[3]Faculty of Medical & Health Sciences, The University of Auckland, Auckland, Auckland, New Zealand
[4]School of Medicine and Dentistry, Griffith University, Gold Coast, Queensland, Australia
[5]Family Health, Gidgee Healing, Mt Isa, Queensland, Australia
[6]Primary Health Care, Kambu Aboriginal and Torres Strait Islander Corporation for Health, Ipswich, Queensland, Australia
[7]Community Child Health, Gold Coast Hospital and Health Service, Southport, Queensland, Australia
[8]Child and Adolescent Unit, Sunshine Coast Hospital and Health Service, Nambour, Queensland, Australia

9Child Development Service, Townsville Hospital and Health Service, Townsville, Queensland, Australia
10Victorian Fetal Alcohol Service, Monash Children's Hospital, Clayton, Victoria, Australia
11Department of Paediatrics, Monash University, Clayton, Victoria, Australia
12Child Health Research Centre, The University of Queensland, Brisbane, Queensland, Australia
13Faculty of Medicine, The University of Queensland, Brisbane, Queensland, Australia
14School of Early Childhood and Inclusive Education, Queensland University of Technology, Brisbane, Queensland, Australia
15Social Marketing, Griffith University, Nathan, Queensland, Australia

**Acknowledgements** We would like to acknowledge the many contributors to the DATAforFASD project. This includes the participating families, who graciously contribute extra time to complete lengthy assessments so that those in the future might get faster access to care. We thank the healthcare practitioners, who have added to their workload by incorporating this protocol into their existing clinical assessments. We thank the partner investigators, chief investigators, associate investigators and steering committee who have guided the conceptualisation and implementation of this project. We would like to thank our Community Advisory Group and remote health practitioners who started the journey to think differently about how to assess FASD so that more children could receive care close to home. We would like to thank Aunty Joan Marshall, Aunty Karen West, Theresa McDonald, Kara Rudken, Kerry Major, Michelle Parker-Tomlin, Keita Hada, Aunty Erica Buttigieg, Alexandrea Hofmann, Kerri O'Conner, Christopher-Lee Wesner and Viliame Sotutu. We are grateful for the support of policy influencers like Zena Martin and Daniel Williamson who have tirelessly advocated for our project. We are grateful for speech pathologists and community members who have helped co-design a language assessment that is best practice for remote First Nations communities. Specifically Jade Bethune, Claire Salter Parry, Bridgette Greathead, Grace Myatt, Bernadette Cantrall, Aunty Shirley Dawson, Taylor Wilson, Chris Doyle and Benton Nemo. This project would not be possible without the support of our research team Heidi Webster, Jade Neuendorf, Amanda Amarrador, Caroline Gaderer, Tia Campbell and Emma Harbeck.

**Contributors** DCS is the principal chief investigator, conceptualised the project, supervised the design and implementation of the research and wrote the first draft of the protocol. EH conceptualised the project, planned the assessment protocols, data collection and analyses and revised the protocol. MZ-G and AJW participated in the project conceptualisation and design of the protocol, advised through membership on the steering committee and contributed to preparation of the manuscript. JB and RSW advised on methodological and analytical aspects of the project, coordinated the health economics data collection and analyses and critically revised the protocol. GS contributed to writing the first draft of the protocol and incorporating coauthor revisions. WL contributed to project conceptualisation, managed project implementation and revised the protocol. CW helped plan data collection, manage project data and storage procedures and contributed to writing the protocol. SH, MP, DS, HT, IM, CH, KH and AC are partner investigators on this project. They coordinate site project implementation, collect data, participated in monthly steering committee meetings and revised the manuscript. NR contributed to conceptualisation of this project, and revised the manuscript. KM, SR, KW and SR-T are chief investigators who participated in steering committee meetings and revised the manuscript.

**Funding** This work was funded by an Australian National Health and Medical Research Council Partnership Grant GNT1170755.

**Competing interests** None declared.

**Patient and public involvement** Patients and/or the public were involved in the design, or conduct, or reporting, or dissemination plans of this research. Refer to the Methods section for further details.

**Patient consent for publication** Not applicable.

**Provenance and peer review** Not commissioned; externally peer reviewed.

**ORCID iDs**
Dianne C Shanley http://orcid.org/0000-0002-3849-075X
Amanda J Wheeler http://orcid.org/0000-0001-9755-674X
Robert S Ware http://orcid.org/0000-0002-6129-6736

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
