## [Reviewer comments · BMJ Open]

ARTICLE DETAILS

TITLE (PROVISIONAL)	Diagnostic Accuracy and economic value of a Tiered Assessment for Fetal Alcohol Spectrum Disorder (DATAforFASD): Protocol
AUTHORS	Shanley, Dianne; Zimmer-Gembeck, Melanie; Wheeler, Amanda; Byrnes, Joshua; Ware, Robert; Liu, Wei; Simcock, Gabrielle; White, Codi; Horton, Sarah; Page, Marjad; Shelton, Doug; Till, Haydn; Mills, Ianthe; Hislop, Carly; Harris, Katrina; Crichton, Alison; Reid, Natasha; Reilly, Sheena; Moritz, Karen; Walsh, K.; Rundle-Thiele, Sharyn; Hawkins, Erinn

VERSION 1 – REVIEW

REVIEWER	Badry, DE University of Calgary, Faculty of Social Work
REVIEW RETURNED	14-Feb-2023

GENERAL COMMENTS	This research protocol is exceptionally well articulated and related to the practical and economic benefit of engaging in FASD diagnosis activity within a primary health care arena. As noted, FASD diagnosis are expensive and occur within specialized clinics. At present no infrastructure exists within primary health care in Australia to undertake these assessments. The protocol identifies a robust series of measures appropriate to the research and has a capable team, clearly identifying the contributions of each member in the protocol. The literature is well utilized and appropriate to current practice and knowledge about FASD in practice and in primary health care. Ethical consent has been obtained including from representatives of the Traditional owners of the land. A critical need exists to embed FASD diagnosis within primary health care and the protocol argues that this is possible and has potential economic costs savings should FASD diagnosis become available through primary health care providers. A tiered assessment approach as indicated has great potential and practically, the plan includes the assessment of 200 children, recognizing that potential limits such as study participation. Further, some limitations may exist in accessing available health care resources, particularly in remote areas. The critical need for FASD diagnosis is highlighted and it is suggested that the potential exists for 55% of the sample to have FASD confirmed. The instrumentation proposed for the research is clearly identified and comprehensive, and identifies the rationale for each index test, all highly suitable for the protocol. The procedures are clearly identified for conducting the research and training for study protocols is included. The inclusion of an economic analysis is a distinct feature of the protocol and recognizes that there are direct and indirect costs associated with FASD diagnosis, including the costs to caregivers, which is rarely identified or recognized in the literature. Data analysis will take place via REDCap which
--

	appropriate and index tests will be compared with diagnostic outcomes in reference tests. Appropriate statistical analysis was identified for this research and results are pending following completion of the research. As a protocol it is evident that significant preparation has been undertaken and this is a sound research protocol. The research will be disseminated via publications and reports to the community. This protocol and paper is exceptionally written with no revisions recommended or required.
--	--

REVIEWER	Hanlon-Dearman, Ana University of Manitoba
REVIEW RETURNED	24-Feb-2023

GENERAL COMMENTS	This is a well conceived, clearly presented protocol. The implications of this work would have wide interest to clinics involved in FASD diagnosis for exactly the reasons outlined in the work. The authors appear to have engaged meaningfully with First Nations/Indigenous groups in the development of the methods. Methodologic evaluation of the RNDA against gold standard battery of assessments for FASD as well as a broad economic cost analysis is appropriate and also well conceived. However, the choice of the RNDA is interesting -- the original research was in Bangladesh with later studies in Guatemala, and I do not see studies that would relate this work to an Australian/first world context. I also did not find more recent study than 2016. The studies comment on the need for further validation and couch comments within a screening paradigm rather than a diagnostic application. I would thus be cautious in interpretation of what appears to be a screening tool compared to a diagnostic battery and the importance of differential diagnosis. Nevertheless, the introductory comments on the costs, waitlists, and resource intensity of FASD assessment are valid and this work will contribute to global considerations on efficient and accurate early diagnosis of this complex disorder. I will look forward to results from this study.
---

REVIEWER	Probert, Adam Public Health Agency of Canada
REVIEW RETURNED	13-Jun-2023

GENERAL COMMENTS	A couple of minor elucidations would help to clarify some points. First a brief explanation for why Brain Structure domains are not included. Second, additional details about who is conducting the caregiver interviews (or whether they will be done at the same time as the usual assessment).
--

VERSION 1 – AUTHOR RESPONSE

Reviewer 1

Comment	Response
This research protocol is exceptionally well articulated and related to the practical and economic benefit of engaging in FASD diagnosis activity within a primary health care	We would like to thank reviewer 1 for their positive feedback.

arena. As noted, FASD diagnosis are expensive and occur within specialized clinics. At present no infrastructure exists within primary health care in Australia to undertake these assessments. The protocol identifies a robust series of measures appropriate to the research and has a capable team, clearly identifying the contributions of each member in the protocol.	
As a protocol it is evident that significant preparation has been undertaken and this is a sound research protocol.	We would like to thank reviewer 1 for noticing the amount of preparation.

Reviewer: 2

Comment	Response
This is a well-conceived, clearly presented protocol. The implications of this work would have wide interest to clinics involved in FASD diagnosis for exactly the reasons outlined in the work. The authors appear to have engaged meaningfully with First Nations/Indigenous groups in the development of the methods. Methodologic evaluation of the RNDA against gold standard battery of assessments for FASD as well as a broad economic cost analysis is appropriate and also well conceived.	We would like to thank reviewer 2 for their positive feedback.
The choice of the RNDA is interesting -- the original research was in Bangladesh with later studies in Guatemala, and I do not see studies that would relate this work to an Australian/first world context. I also did not find more recent study than 2016. The studies comment on the need for further validation and couch comments within a screening paradigm rather than a diagnostic application.	Our team have administered the RNDA to around 900 Australian children, including remote and First Nations children. Preliminary evidence suggests the RNDA already functions well as a screening tool. We have included a recent citation of our use of the RNDA as a developmental screener in preventative health checks in the Australian context on page 11. Additional studies documenting reliability and validity in an Australian context are being prepared. The current diagnostic accuracy study will help us to further understand the sensitivity, specificity and diagnostic capacity of the RNDA in an Australian context.

Reviewer: 3

Comment	Response
A brief explanation for why Brain Structure domains are not included.	A sentence has been added to page 7: The reference test for the brain structure domain is already rapid (i.e., history of seizures, head circumference or brain imaging if available). Therefore, an index test for this domain will not be administered. Information for this domain will still be collected for diagnostic purposes.

Additional details about who is conducting the caregiver interviews (or whether they will be done at the same time as the usual assessment)	A sentence has been added to the methods section with additional details about who is conducting the caregiver interviews and when: Enrolled families and healthcare practitioners will complete surveys with one member from the research team that capture prospective cost information at the completion of testing.
---	---

Thank-you again for reviewing our manuscript. We look forward to the next steps in the publication journey.

VERSION 2 – REVIEW

REVIEWER	Hanlon-Dearman, Ana University of Manitoba
REVIEW RETURNED	04-Jul-2023

GENERAL COMMENTS	Very clearly organized research. This work is of international interest and value to multidisciplinary diagnostic clinics faced with the challenges of long and complex assessments of brain domains as required by the various guidelines. By comparing index tests to gold standard appropriately there may be system benefits to individuals and families as well as cost savings as you have outlined. Will look forward to your results with interest.
---

REVIEWER	Probert, Adam Public Health Agency of Canada
REVIEW RETURNED	14-Jul-2023

GENERAL COMMENTS	This is a well developed and written proposal. I appreciate the time and effort spent in revising . There is a critical need for this, and similar studies and we look forward to the results.
---